# Disparate temperature-dependent virus–host dynamics for SARS-CoV-2 and SARS-CoV in the human respiratory epithelium

Philip V'kovski[1,2⊘], Mitra Gultom[1,2,3,4⊘], Jenna N. Kelly[1,2⊘], Silvio Steiner[1,2,3⊘],
Julie Russeil[5], Bastien Mangeat[6], Elisa Cora[6], Joern Pezoldt[5], Melle Holwerda[1,2,3,4],
Annika Kratzel[1,2,3], Laura Laloli[3,4], Manon Wider[4], Jasmine Portmann[1,2], Thao Tran[1,2,3],
Nadine Ebert[1,2], Hanspeter Stalder[1,2], Rune Hartmann[7], Vincent Gardeux[5,8],
Daniel Alpern[5,8], Bart Deplancke[5,8], Volker Thiel[1,2], Ronald Dijkman[1,2,4]*

1 Institute of Virology and Immunology (IVI), Bern, Switzerland, 2 Department of Infectious Diseases and Pathobiology, Vetsuisse Faculty, University of Bern, Bern, Switzerland, 3 Graduate School for Biomedical Science, University of Bern, Bern, Switzerland, 4 Institute for Infectious Diseases, University of Bern, Bern, Switzerland, 5 Institute of Bioengineering, School of Life Sciences, École Polytechnique Fédérale de Lausanne (EPFL), Lausanne, Switzerland, 6 Gene Expression Core Facility (GECF), School of Life Sciences, École Polytechnique Fédérale de Lausanne (EPFL), Lausanne, Switzerland, 7 Department of Molecular Biology and Genetics, Aarhus University, Aarhus, Denmark, 8 Swiss Institute of Bioinformatics, Lausanne, Switzerland

⊘ These authors contributed equally to this work.
* ronald.dijkman@ifik.unibe.ch

**Data Availability Statement:** Transcriptome data has been deposited in the Arrayexpress open-access public repository from the European Bioinformatics Institute (EMBL-EBI) under E-

## Abstract

Since its emergence in December 2019, Severe Acute Respiratory Syndrome Coronavirus 2 (SARS-CoV-2) has spread globally and become a major public health burden. Despite its close phylogenetic relationship to SARS-CoV, SARS-CoV-2 exhibits increased human-to-human transmission dynamics, likely due to efficient early replication in the upper respiratory epithelium of infected individuals. Since different temperatures encountered in the human upper and lower respiratory tract (33°C and 37°C, respectively) have been shown to affect the replication kinetics of several respiratory viruses, as well as host innate immune response dynamics, we investigated the impact of temperature on SARS-CoV-2 and SARS-CoV infection using the primary human airway epithelial cell culture model. SARS-CoV-2, in contrast to SARS-CoV, replicated to higher titers when infections were performed at 33°C rather than 37°C. Although both viruses were highly sensitive to type I and type III interferon pretreatment, a detailed time-resolved transcriptome analysis revealed temperature-dependent interferon and pro-inflammatory responses induced by SARS-CoV-2 that were inversely proportional to its replication efficiency at 33°C or 37°C. These data provide crucial insight on pivotal virus–host interaction dynamics and are in line with characteristic clinical features of SARS-CoV-2 and SARS-CoV, as well as their respective transmission efficiencies.

MTAB-9781, and scripts used for analysis and figure generation will be available via our Github repository (https://github.com/IFIK-virology).

**Funding:** The authors received funding from the following sources: European Commission (Marie Sklodowska-Curie Innovative Training Network "HONOURS"; grant agreement No 721367) to VT and RD (https://cordis.europa.eu/project/id/721367), The Swiss National Science Foundation (SNSF) grants 179260 to RD (http://p3.snf.ch/project-179260), 173085 to VT (http://p3.snf.ch/project-173085), 31CA30_196644 to VT and RD, (http://p3.snf.ch/project-196644), National Center of Competence in Research (NCCR) on RNA and Disease to VT (https://nccr-rna-and-disease.ch/), German Federal Ministry of Education and Research (BMBF), grant RAPID (01KI1723A) to VT and RD. The funders had no role in study design, data collection and analysis, decision to publish, or preparation of the manuscript.

**Competing interests:** The authors have declared that no competing interests exist.

**Abbreviations:** ACE2, angiotensin-converting enzyme 2; ALI, air-liquid interface; BRB-seq, Bulk RNA Barcoding and sequencing; COVID-19, Coronavirus Disease 2019; DAPI, 4′,6-diamidino-2-phenylindole; DE, differentially expressed; FBS, fetal bovine serum; FC, fold change; FDR, false discovery rate; GSEA, gene set enrichment analysis; hAEC, human airway epithelial cell; HBSS, Hanks balanced salt solution; hpi, hours post infection; IFN, interferon; IFIT, interferon-induced proteins with tetratricopeptide repeats; ISG, IFN-stimulated gene; LRT, Likelihood Ratio Test; MOI, multiplicity of infection; OASL, 2′-5′-oligoadenylate synthetase-like; PFU, plaque-forming unit; PRR, pattern recognition receptor; SARS-CoV-2, Severe Acute Respiratory Syndrome Coronavirus 2; scRNA-seq, single-cell RNA-sequencing; UMAP, Uniform Manifold Approximation and Projection; UMI, unique molecule identifier; VST, variance-stabilizing transformation; VTM, virus transport medium.

## Introduction

The zoonotic coronavirus Severe Acute Respiratory Syndrome Coronavirus 2 (SARS-CoV-2) first emerged in Wuhan, Hubei Province, China, in December 2019 and was soon recognized as the etiological agent of Coronavirus Disease 2019 (COVID-19). To date, the COVID-19 pandemic has resulted in over 115 million laboratory-confirmed cases worldwide, including more than 2.5 million deaths [1–4]. Interestingly, SARS-CoV-2 has a close phylogenetic relationship with SARS-CoV, another coronavirus that emerged in 2002/2003 and led to over 8,000 confirmed cases and 800 deaths [5]. SARS-CoV-2 differs from SARS-CoV by only 380 amino acids and retains a high level of conservation in known receptor-binding motifs that interact with the human receptor angiotensin-converting enzyme 2 (ACE2) [6]. Moreover, although the cell surface receptor ACE2 and the serine protease TMPRSS2 have been shown to serve as entry determinants for both SARS-CoV and SARS-CoV-2 [2,7–9], an accumulating body of evidence shows that the 2 viruses exhibit distinct human-to-human transmission dynamics and follow different clinical courses of infection. These differences between SARS-CoV and SARS-CoV-2 strongly suggest the presence of disparate virus–host dynamics during viral infection in the human respiratory epithelium [10–15].

The human conductive respiratory tract is lined by a pseudostratified, ciliated, and columnar epithelium that contains mucin-producing goblet cells and represents a crucial barrier to constrain invading pathogens. The anatomical distance between the upper and lower respiratory conductive tract and their different ambient temperatures (32 to 33°C and 37°C, respectively [16,17]) have previously been shown to influence the replication kinetics of diverse respiratory viruses, such as rhinoviruses, influenza viruses, and coronaviruses [18–22]. Moreover, the anatomical disparity in ambient temperature also affects virus–host immune response dynamics and, thus, potential human-to-human transmission dynamics [23]. Interestingly, SARS-CoV-2 has been detected earlier after infection and more abundantly than SARS-CoV in upper respiratory tissues of infected patients [10,13,14,24,25], suggesting that transmission kinetics and host innate immune response dynamics in the infected tissues might differ between SARS-CoV and SARS-CoV-2 infections.

Here, we employed the human airway epithelial cell (hAEC) culture model to investigate the influence of different incubation temperatures on the viral replication kinetics and host immune response dynamics of both SARS-CoV and SARS-CoV-2 infections. Our study revealed that SARS-CoV-2 replication improved in hAEC cultures incubated at 33°C rather than 37°C and that higher infectious titers were recovered from cultures infected with SARS-CoV-2 than hAEC cultures infected with SARS-CoV at 33°C. Both SARS-CoV and SARS-CoV-2 replicated equally efficiently in hAEC cultures at 37°C. Pretreatment of hAEC cultures with exogenous type I and III interferon (IFN) at different temperatures showed that SARS-CoV-2 and SARS-CoV are highly sensitive to both type I and III IFN, thereby exemplifying the relevance of early IFN signaling and innate immune responses to restrict viral infection. Importantly, a detailed temporal transcriptome analysis of infected hAEC cultures corroborated initial findings and uncovered characteristic innate immune response gene signatures inversely correlating with the viral replication efficiency of SARS-CoV-2 at different ambient temperatures. Altogether, these results provide an in-depth fundamental insight on the virus–host innate immune response dynamics of SARS-CoV-2 and the closely phylogenetically related SARS-CoV in the respiratory epithelium and concur the clinical characteristics and transmission efficiencies of both viruses.

## Results

### Replication kinetics of SARS-CoV-2 and SARS-CoV at 33°C and 37°C

hAEC cultures are a well-characterized in vitro model that morphologically and functionally recapitulate the epithelial lining of the human respiratory tract in vivo. hAEC cultures from 7

different human donors were inoculated with either SARS-CoV-2/München-1.1/2020/929 or SARS-CoV Frankfurt-1 isolates using a multiplicity of infection (MOI) of 0.1. Infected hAEC cultures were incubated at either 33˚C or 37˚C throughout the experiment to assess the influence of the temperature variations that occur along the human respiratory tract and to model-associated virus–host interaction dynamics. The polarity of viral progeny release was monitored by collecting apical washes and basolateral medium with 24-hour intervals for a period of 96 hours. At 37˚C, SARS-CoV and SARS-CoV-2 replicated to similar titers over the course of the infection (Fig 1A and 1B). Interestingly, when assessing viral replication at 33˚C

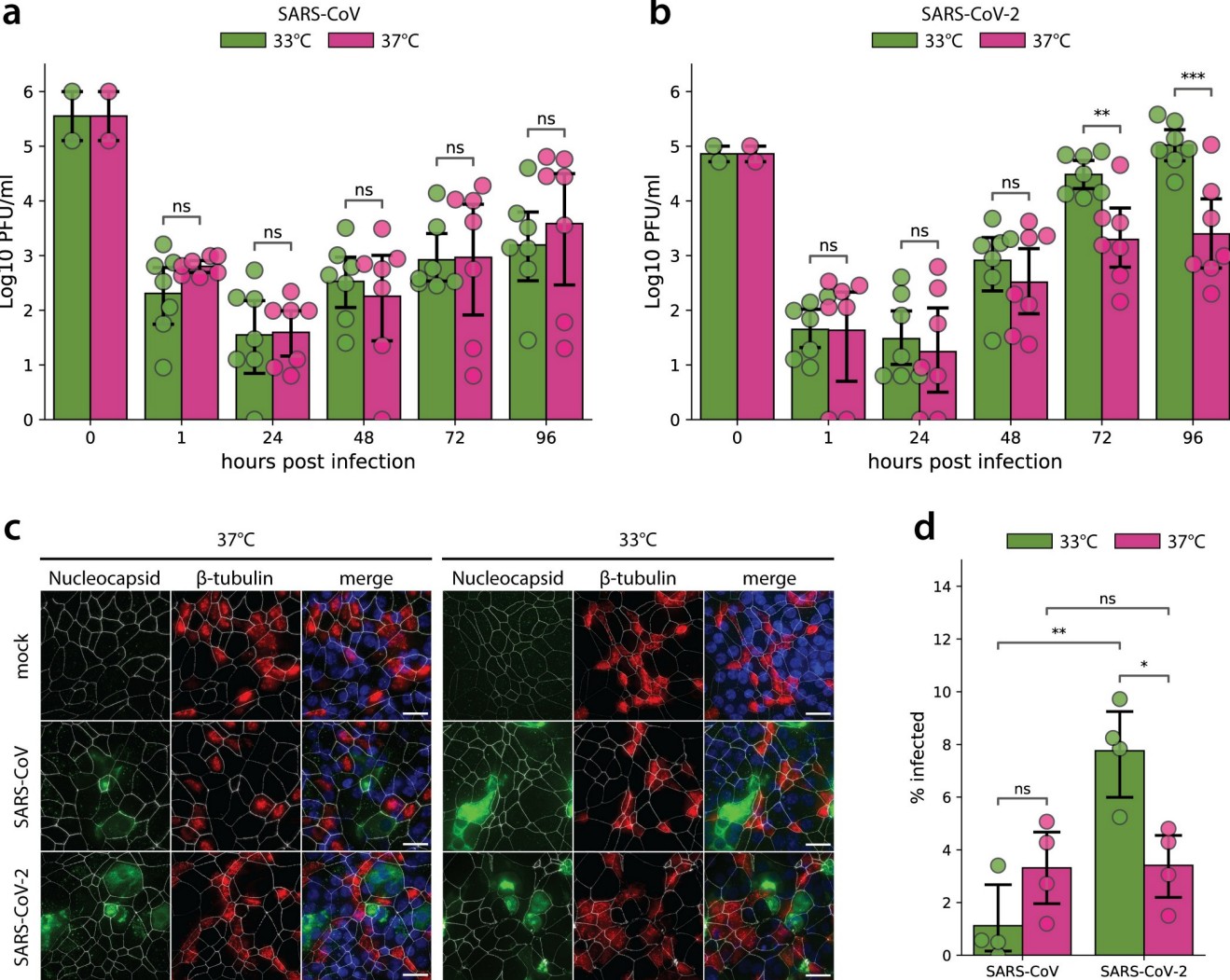

**Fig 1. SARS-CoV and SARS-CoV-2 replication kinetics in hAEC cultures.** Well-differentiated hAEC cultures were infected with SARS-CoV (**a**) and SARS-CoV-2 (**b**) using 30,000 PFU or remained uninfected (mock) and were incubated at 37˚C or 33˚C. Inoculated virus was removed at 1 hpi and the apical side was washed. Cultures were further incubated at the indicated temperature. At the indicated time post infection, apical virus release was assessed by plaque titration (**a, b**). Data represent the mean ± 95% CI of hAEC cultures from 7 different human donors. Individual points represent the average of 2 technical replicates. Values at 0 hpi indicate the titer of the inoculum used to infect the hAEC cultures, and values at 1 hpi indicate the remaining titer after the third wash. The *p*-values were computed by using two-sided paired sample *t* tests. At 96 hpi, hAEC cultures were fixed and processed for immunofluorescence analysis using antibodies against SARS-CoV Nucleocapsid protein (green), β-tubulin (cilia, red), ZO-1 (tight junctions, white), and DAPI (blue) (**c**). Representative z-projections of one donor are shown. Scale bar, 20 μm. Infected cells were quantified after segmentation of individual cells based on the ZO-1 staining and measuring the mean intensity of the nucleocapsid protein staining (**d**). Data represent the mean ± 95% CI of multiple images acquired from hAEC cultures derived from 4 different human donors. On average, more than $10^4$ cells per donor and per condition were analyzed. The data underlying this figure are found in S1 Data. hAEC, human airway epithelial cell; hpi, hours post infection; PFU, plaque-forming unit; SARS-CoV, Severe Acute Respiratory Syndrome Coronavirus; SARS-CoV-2, Severe Acute Respiratory Syndrome Coronavirus 2.

rather than at 37˚C, it was apparent that SARS-CoV-2 infection resulted in 10-fold higher titers released in the apical compartment between 72 and 96 hours post infection (hpi). In contrast, SARS-CoV replication at 33˚C remained similar to replication at 37˚C and showed no significant differences over the entire course of infection (Fig 1A and 1B). Since the directionality of viral progeny release is crucial for subsequent virus spread and overall disease outcome, basolateral release of SARS-CoV and SARS-CoV-2 were also assessed. Similar to what we and others have observed previously for all other human coronaviruses, SARS-CoV and SARS-CoV-2 were predominantly released to the luminal surface (S1A and S1B Fig) [26,27].

To assess whether the observed different temperature-dependent replication efficiencies are a result of the number of cells infected by SARS-CoV or SARS-CoV-2, hAEC cultures were fixed at 96 hpi and processed for immunofluorescence analysis using antibodies directed against the SARS-CoV Nucleocapsid protein. Additionally, to discern potential preferential virus tropism to a distinct cell type, characteristic markers of the hAEC cultures' architecture, such as the intercellular tight junctions (ZO-1) and cilia (β-tubulin), were also included. Microscopy investigations and automated image quantification revealed that concurrent with the more efficient replication kinetics of SARS-CoV-2 at 33˚C, the fraction of SARS-CoV-2-infected cells increased significantly at 33˚C compared to 37˚C and to the fraction of SARS-CoV-infected cells, which remained similar at 33˚C and 37˚C. Both viruses displayed a comparable fraction of infected cells at 37˚C (Fig 1C and 1D). Notably, at 72 and 96 hpi, the majority of SARS-CoV and SARS-CoV-2 antigen positive cells were not costained by the β-tubulin marker and were therefore qualified as nonciliated cell (Figs 1C and 2C). We previously observed that SARS-CoV infects both nonciliated and ciliated cell populations [26]. However, given that other reports show that SARS-CoV primarily targets ciliated cells [28], we analyzed the localization of the entry receptor, ACE2, and β-tubulin markers by microscopy. Immunofluorescence analysis revealed that ACE2 is expressed in both ciliated and nonciliated cell populations in uninfected hAEC cultures (S2A Fig). In line with this, the analysis of mRNA expression in noninfected hAEC cultures using single-cell RNA-sequencing (scRNA-seq) confirmed that both *ACE2* and *TMPRSS2* mRNA are found in both secretory and ciliated cell populations (S2B–S2D Fig) [29]. Combined, these results demonstrate that despite their shared requirement on ACE2 and TMPRSS2 for entry into host cells, SARS-CoV and SARS-CoV-2 display strong temperature-dependent variation in replication kinetics in hAEC cultures, suggestive of host determinants intervening during post-entry stages of the viral life cycle. Importantly, the significantly enhanced replication of SARS-CoV-2 at 33˚C likely supports the increased replication in the upper respiratory tract and transmissibility of SARS-CoV-2 compared to SARS-CoV.

## Sensitivity of SARS-CoV-2 and SARS-CoV to IFN

The amount of viral progeny released from infected cells is representative of the dynamic interplay between viral replication and its inhibition by cellular defense mechanisms, such as by different types of interferon-stimulated genes (ISGs). To examine whether the induction of ISGs upon interferon stimulation differentially affects SARS-CoV and SARS-CoV-2 replication, hAEC cultures were pretreated with 50 IU or 5 IU of exogenous type I interferon (IFN-αA/D) and 50 or 5 ng of type III interferon (IFN-3), for 18 hours prior to infection, at either 33˚C or 37˚C. Hereafter, the hAEC culture medium was replaced with IFN-free medium, and cells were infected with SARS-CoV and SARS-CoV-2 at an MOI of 0.1, at 33˚C or 37˚C, for 72 hours. The titration of apically released virus revealed that the replication of both SARS-CoV-2 and SARS-CoV is severely restricted upon pretreatment with 50 IU of type I or 50 ng of type III IFN at either 33˚C or 37˚C (Fig 2A). Although, similar to previously reported observations

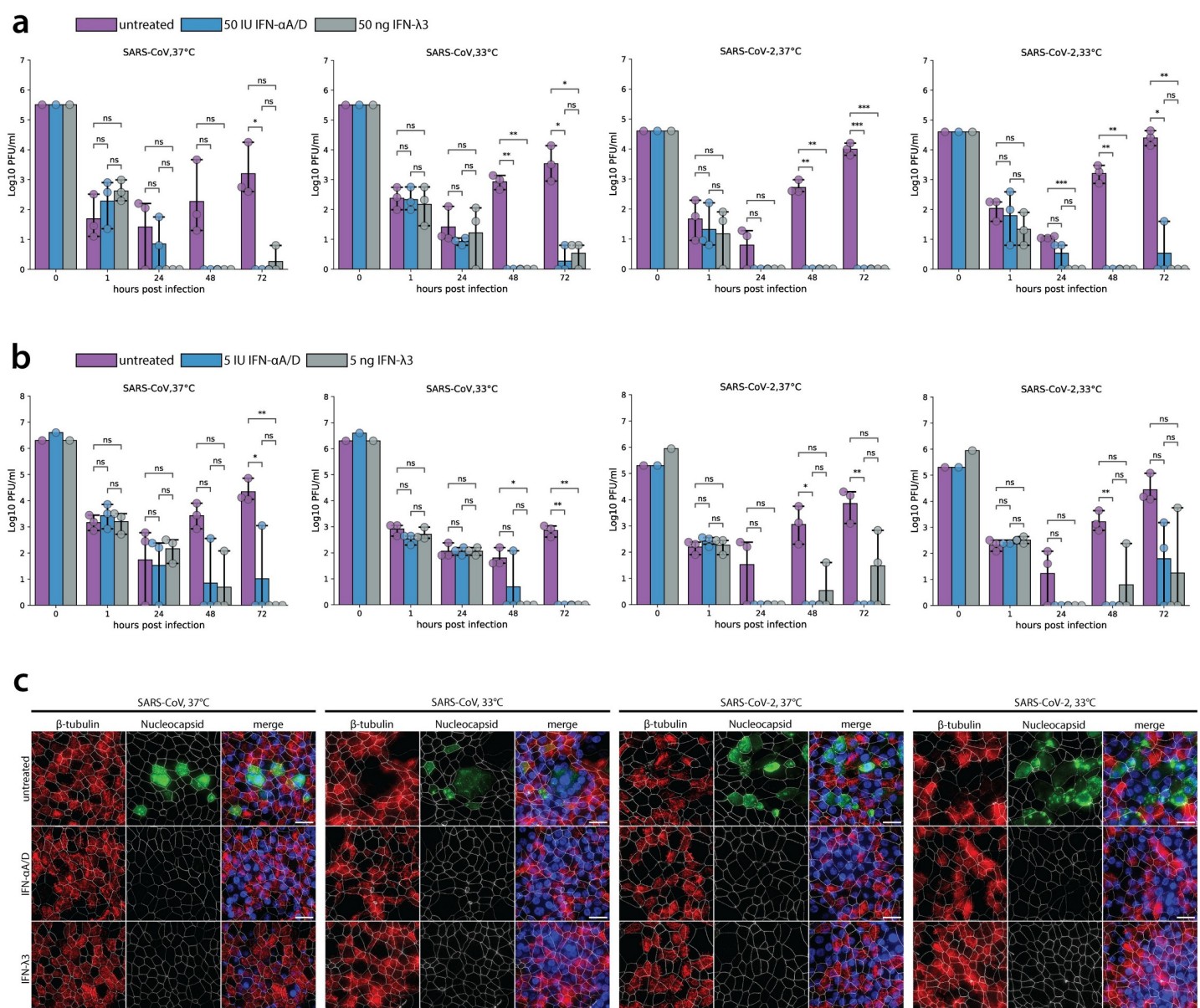

**Fig 2. SARS-CoV and SARS-CoV replication upon IFN-I and IFN-III pretreatment.** hAEC cultures were treated from the basolateral side with recombinant universal type I IFN (50 IU or 5 IU) or recombinant IFN-λ3 (50 ng or 5 ng) for 18 hours. Before infection, medium was removed and replaced with IFN-free medium, and hAEC cultures were infected with SARS-CoV and SARS-CoV-2 using 30,000 PFU and were incubated at 37˚C or 33˚C. Inoculated virus was removed at 1 hpi, and the apical side was washed. Cultures were further incubated at the indicated temperature. At the indicated time, apical virus release was assessed by plaque titration (**a, b**). Data represent the mean ± 95% CI of hAEC cultures from 3 different human donors. Individual points represent 1 (b) or the average of 2 technical replicates (a). Values at 0 hpi indicate the titer of the inoculum used to infect the hAEC cultures, and values at 1 hpi indicate the remaining titer after the third wash. The *p*-values were computed by using two-sided paired sample *t* tests. The data underlying this figure are found in S1 Data. At 72 hpi, hAEC cultures pretreated with 50 IU type I IFN or 50 ng type III IFN were fixed and processed for immunofluorescence analysis using antibodies against SARS-CoV Nucleocapsid protein (green), β-tubulin (cilia, red), ZO-1 (tight junctions, white), and DAPI (blue) (**c**). Representative z-projections of 1 donor are shown. Scale bar, 20 μm. hAEC, human airway epithelial cell; hpi, hours post infection; IFN, interferon; PFU, plaque-forming unit; SARS-CoV, Severe Acute Respiratory Syndrome Coronavirus; SARS-CoV-2, Severe Acute Respiratory Syndrome Coronavirus 2.

[26], SARS-CoV seemed less sensitive to type I IFN than to type III IFN pretreatment at 37˚C (Fig 2A, 24 hpi). However, these differences were not statistically significant, which was further confirmed by pretreatment with 5 IU of type I IFN or 5 ng of type III IFN (Fig 2B). Consistently, SARS-CoV displayed a similar sensitivity to type I and III IFN pretreatments at 33˚C

(Fig 2A and 2B). In line with these results, SARS-CoV-2 was equally sensitive to both doses of type I or III IFN pretreatments at either 33°C or 37°C (Fig 2A and 2B). Notably, restriction of SARS-CoV-2 upon pretreatment with 5 IU of type I and 5 ng of type III IFNs was moderately relieved at 72 hpi (Fig 2B).

The reduction of viral progeny titers in both IFN pretreatment conditions were corroborated by immunofluorescence analysis at 72 hpi. Viral antigens were no longer detected upon immunostaining of the type I or III IFN-pretreated hAEC cultures with anti-SARS-CoV Nucleocapsid protein antibodies (Fig 2C). Altogether, these results suggest that the viral replication kinetics of both SARS-CoV and SARS-CoV-2 in the upper and lower airways are heavily dependent on innate immune responses elicited by type I and III IFN and that a potent IFN response can efficiently restrict viral replication of SARS-CoV-2 in primary well-differentiated hAEC cultures.

## Disparate host responses to SARS-CoV and SARS-CoV-2 in hAEC cultures

Given the striking impact of temperature on SARS-CoV-2 replication kinetics (Fig 1) and the reduction in viral load following IFN pretreatment (Fig 2), we next sought to investigate how differences in temperature may influence the host transcriptional response to SARS-CoV and SARS-CoV-2. To this end, cellular RNA was extracted from hAEC cultures that were infected with either SARS-CoV or SARS-CoV-2 (MOI 0.1), as well as from uninfected hAEC cultures, at 24, 48, 72, and 96 hpi (Fig 1). Samples were then sequenced for transcriptome analysis using the Bulk RNA Barcoding and sequencing (BRB-seq) protocol to a raw sequencing depth of 12 million reads per sample [30]. This sequencing depth was used to ensure that all samples were sequenced to saturation and that genes with lower expression levels were included in the analysis, as illustrated in S3 Fig. Inspection of the abundance of SARS-CoV and SARS-CoV-2 virus-specific reads demonstrated that their levels of expression were consistent with their viral replication kinetics at 33°C and 37°C described in Fig 1 (S4 Fig). Downstream analysis of differentially expressed (DE) host genes across datasets was performed using 3 distinct approaches to (i) identify global differences between SARS-CoV and SARS-CoV-2 infection influenced by temperature alone, as well as (ii) perform individual pairwise comparisons between each virus and its uninfected counterpart at different time points and temperatures, and (iii) perform temporal analysis to identify DE genes that change over time.

For the first DE gene analysis, combined data from the 7 biological donors was normalized and segregated by temperature alone to uncover global changes in gene expression in SARS-CoV and SARS-CoV-2 infected hAEC cultures relative to uninfected hAEC cultures at 33°C and 37°C. This approach identified a total of 126 DE genes for SARS-CoV infections at 33°C, 2 DE genes for SARS-CoV infections at 37°C, 161 DE genes for SARS-CoV-2 infections at 33°C, and 82 DE genes for SARS-CoV-2 infections at 37°C (Log$_2$FC $\geq$ 1.5, FDR $\leq$ 0.1), represented by a total of 276 unique genes (S1 Table). Comparison of the DE genes identified at different temperatures for each virus revealed several gene clusters that were specific to either SARS-CoV or SARS-CoV-2 infection, and others that were shared among distinct conditions (Fig 3A). Pathway enrichment analysis of the individual gene clusters revealed that the 43 DE genes shared between SARS-CoV and SARS-CoV-2 infections at 33°C were predominantly associated with eukaryotic mRNA translation pathways, whereas the specific genes for SARS-CoV infection at 33°C were mostly related to chemokine signaling pathways (Fig 3B and S2 Table). In contrast, DE genes identified for SARS-CoV-2 infections at both 33°C and 37°C were mainly associated with the host antiviral response (Fig 3B).

To identify DE genes that may be important at specific time points, we next segregated the combined normalized data by both temperature and time and performed pairwise

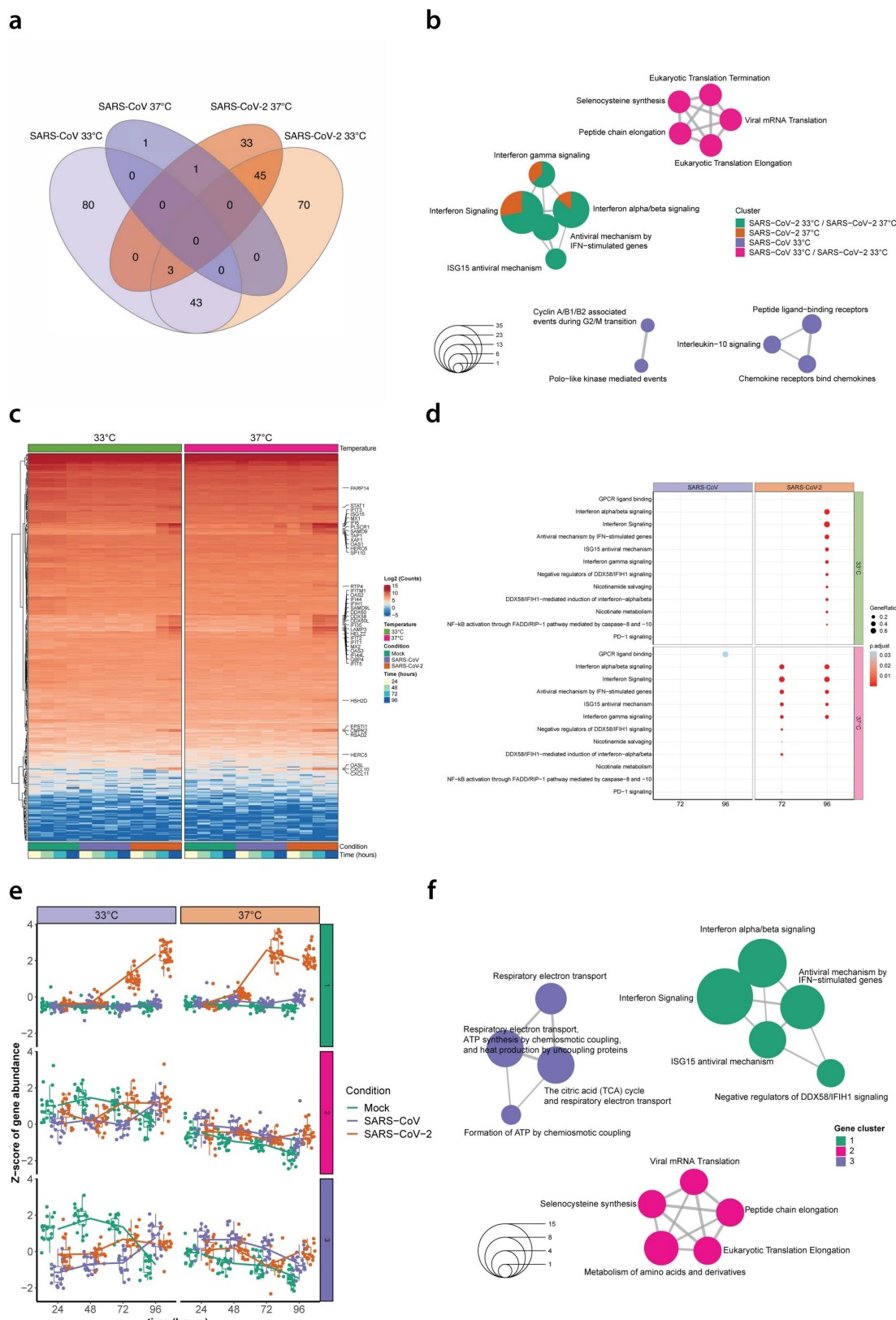

**Fig 3. Temperature-dependent host transcriptional response in SARS-CoV and SARS-CoV-2 infected hAEC cultures.** Venn diagram showing the overlap among DE genes identified in SARS-CoV (purple) and SARS-CoV-2 (orange) infected hAEC

cultures at either 33˚C or 37˚C (**a**). Enrichment map illustrating the pathway enrichment analysis results for the DE gene clusters identified in SARS-CoV and SARS-CoV-2 infected hAEC cultures at either 33˚C or 37˚C. Circle size represents the number of genes associated with a given pathway (**b**). Hierarchical cluster analysis of DE genes identified in SARS-CoV and SARS-CoV-2 infected hAEC cultures at either 33˚C or 37˚C compared to uninfected hAEC cultures (Mock). Expression levels for individual DE genes are shown in rows as the $\log_2$ mean normalized counts for all 7 human donors stratified by condition, temperature, and hpi (columns; representative colors shown in legends). The 40 of 45 DE genes unique to SARS-CoV-2 at both 33˚C and 37˚C are shown on the right (y-axis) (**c**). Dotplot illustrating pathway enrichment analysis performed on the 16 distinct DE gene profiles. Significantly enriched pathways for SARS-CoV and SARS-CoV-2 are shown for both 33˚C and 37˚C incubation temperatures at 72 and 96 hpi. Dots were adjusted in size and color to illustrate the gene ratio and adjusted *p*-value (<0.05) for a given pathway, respectively (**d**). Boxplot graphs showing the distribution of the Z-score abundance from DE genes associated with cluster 1, 2, and 3 at different hpi for Mock (green), SARS-CoV (purple), and SARS-CoV-2 (orange) for both 33˚C and 37˚C. The overall mean Z-score abundance over time is for each condition indicated with a solid line (**e**). Enrichment map illustrating the pathway enrichment analysis on the temporal DE genes identified in SARS-CoV and SARS-CoV-2 infected and uninfected (Mock) hAEC cultures at either 33˚C or 37˚C. Circle size represents the number of genes associated with a given pathway (**f**). DE, differentially expressed; hAEC, human airway epithelial cell; hpi, hours post infection; SARS-CoV, Severe Acute Respiratory Syndrome Coronavirus; SARS-CoV-2, Severe Acute Respiratory Syndrome Coronavirus 2.

comparisons between SARS-CoV-2 or SARS-CoV samples and their corresponding uninfected samples at each temperature and time point. Interestingly, this analysis uncovered that irrespective of temperature and time the overlap between the host response induced by each virus was relatively low and that most DE genes for SARS-CoV-2 were present at 48 and 96 hpi for 33˚C and 72 and 96 hpi for 37˚C ($\log_2$FC ≥ 1.5, FDR ≤ 0.1), while for SARS-CoV, the majority of DE genes were identified at 48 hpi for 33˚C, representing 226 of the 401 unique genes identified (S5A–S5D Fig and S3 Table). A similar trend was observed when pairwise comparisons were performed directly between SARS-CoV-2 and SARS-CoV samples. Notably, these findings revealed that the most contrasting differences between SARS-CoV and SARS-CoV-2 occurred at 96 hpi for 33˚C, whereas they occurred at 72 hpi for 37˚C (S5E and S5F Fig and S4 Table). To further explore these results, we performed hierarchical clustering with the 401 unique identified DE genes and annotated 40 of the 45 DE genes that were also found to be specific to SARS-CoV-2 infection at both 33˚C and 37˚C from the global analysis (Fig 3A and 3C). Furthermore, to establish whether certain biological pathways were significantly modulated over time at the different temperatures, pathway enrichment analysis was performed on all unique DE gene clusters detected in the pairwise comparisons for SARS-CoV or SARS-CoV-2. Overall, these results revealed a distinct temperature-dependent host response for SARS-CoV-2 at 72 and 96 hpi, including the enrichment of diverse IFN and antiviral signaling genes and pathways (Fig 3C and 3D). Notably, none of these antiviral genes or pathways were significantly enriched during SARS-CoV infection.

Following the pairwise clustering analysis, which directly compares SARS-CoV or SARS-CoV-2 infections to uninfected hAEC cultures at each individual time point, we also performed a global temporal DE analysis, which is tailored to highlight significant gene expression changes over time. This analysis identified a total of 98 DE genes (FDR ≤ 0.1), representing 70 up-regulated and 28 down-regulated genes, that fell into 3 distinct hierarchical gene clusters (Fig 3E and S5 Table). Most DE genes in clusters 1, 2, and 3 appeared to be associated with IFN-signaling, eukaryotic mRNA translation, or respiratory electron transport pathways, respectively (Fig 3F). Closer inspection of the DE gene expression level changes over time revealed that only genes associated with cluster 1 (IFN-signaling) showed a clear temporal and temperature-dependent expression pattern. Interestingly, for SARS-CoV-2 infections, expression of these genes increased among all 7 donors as early as 48 hpi at 37˚C, but not at 33˚C (Fig 3E). This finding was less apparent for SARS-CoV, which showed only a marginal increase in expression for the 29 IFN-signaling associated DE genes at 96 hpi for 37˚C (Fig 3E).

Together, these results illustrate that the closely related viruses SARS-CoV and SARS-CoV-2 induce disparate and temperature-dependent host responses in hAEC cultures that vary over

time and correlate with their replication phenotypes at 33˚C and 37˚C. Moreover, in contrast to SARS-CoV, SARS-CoV-2 triggered a pronounced antiviral and pro-inflammatory response in primary human airway epithelial cell cultures. These responses occurred earlier and were more strongly induced at 37˚C compared to 33˚C and coincided with the increased replication of SARS-CoV-2 at temperatures corresponding to the upper respiratory epithelium (33˚C).

## Innate immune responses to SARS-CoV and SARS-CoV-2 in hAEC cultures

To gain further insight into the dynamics of the innate immune responses to SARS-CoV and SARS-CoV-2 infection, we examined the expression levels of the 29 DE genes found in cluster 1 of our temporal analysis that were associated with IFN signaling pathways (cluster 1 in Fig 3E and 3F). These DE genes were plotted along with the 401 genes identified in our previous pairwise comparisons of SARS-CoV and SARS-CoV-2 at both 33˚C and 37˚C (S6 Fig). This highlighted that *CXCL10* and *CXCL11*, chemokines responsible for immune cell recruitment to the site of infection from the bloodstream, were among the core group of 29 DE genes and could be detected at both temperatures, albeit up-regulation was stronger at 37˚C compared to 33˚C. Conversely, no up-regulation of the canonical pro-inflammatory genes such as *TNF*, *IL-11*, *IL-18*, *IL-6*, and *IL1B* was observed during SARS-CoV-2 infections of hAEC cultures (S7A and S7B Fig). The pattern recognition receptor (PRR) *RIG-I/DDX58* and interferon-inducible 2′-5′-oligoadenylate synthetase-like protein (*OASL*), and 3 members of the interferon-induced proteins with tetratricopeptide repeats (IFITs) family were also identified as some of the most prominent DE genes that showed an earlier and higher expression at 37˚C during SARS-CoV-2 infections (Fig 3C and S6 Fig). In addition, gene set enrichment analysis (GSEA) revealed a high degree of interconnectivity among DE genes associated with the diverse enriched IFN and antiviral signaling pathways, further highlighting the importance of the host antiviral response during SARS-CoV-2 infection (Fig 4A).

We previously observed, in the context of influenza AH1N1pdm09 virus, that only a small fraction of infected cells produces IFN during infection [31]. Since the majority of up-regulated DE genes during SARS-CoV-2 infection were genes induced downstream of the IFN pathway, we next assessed the expression of individual type I and III IFN genes over time. Interestingly, although only *IFNL1* and *IFNB1* were significantly up-regulated in the global analysis (S1 Table), the expression levels of *IFNL2* and *IFNL3* (Fig 4B) also followed a similar temperature-dependent pattern as the ISGs highlighted in Fig 3C. In contrast, and in agreement with previous results (Fig 3A and 3E), SARS-CoV infection did not induce type I or III IFNs at either 33˚C or 37˚C (Fig 4B). Notably, we confirmed the expression of both type I and III IFN receptors (*IFNAR1*, *IFNAR2*, *IFNLR1*, and *IL10RB*) in multiple cell types of noninfected hAEC cultures using scRNA-seq (S2E–S2H Fig). Together, these results show that SARS-CoV infection triggers a relatively mild induction of IFN in hAEC cultures, whereas SARS-CoV-2 infection leads to stronger induction of multiple IFNs that is dominated by type III IFNs and dependent on temperature. However, whether more potent innate immune activation restricts SARS-CoV-2 replication at 37˚C, or whether a distinct virus–host interplay favors SARS-CoV-2 replication at 33˚C awaits to be formally determined. Nevertheless, the more substantial IFN-driven innate immune signaling observed at 37˚C rather than 33˚C coincides with the more efficient replication of SARS-CoV-2 at 33˚C.

## Discussion

In the current study, we demonstrate that the ambient temperatures reminiscent of the conditions in the upper and lower respiratory tract have a profound influence on both

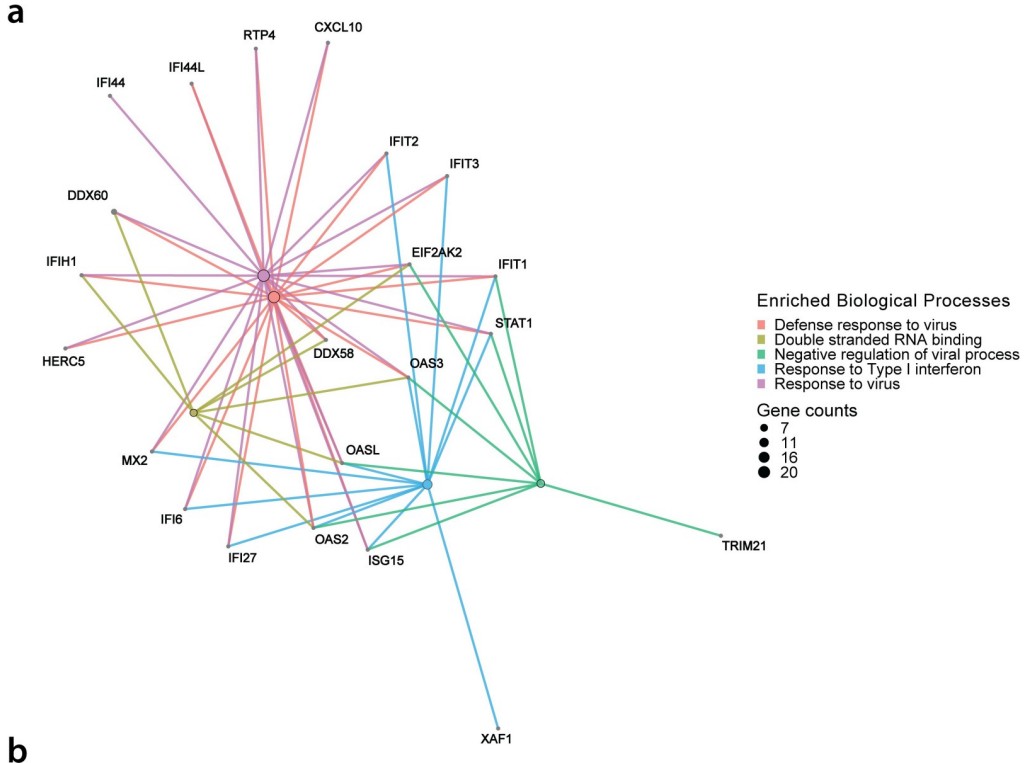

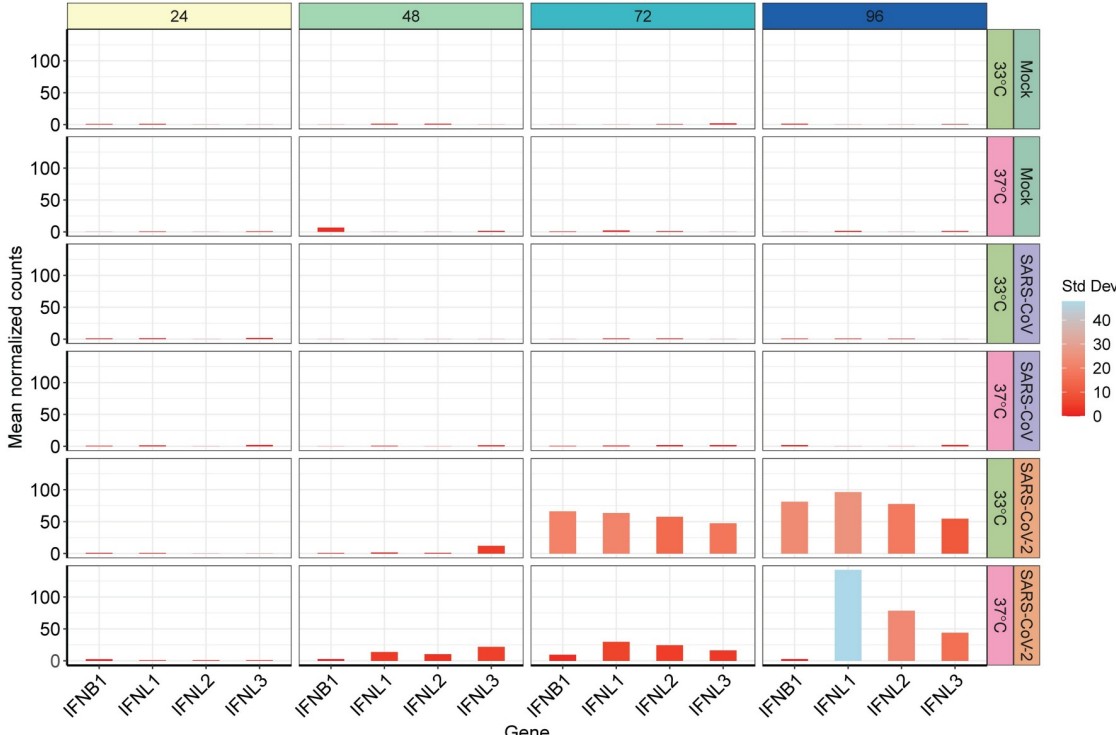

**Fig 4. Innate immune response in SARS-CoV and SARS-CoV-2 infected hAEC cultures.** Gene-concept network plot illustrating the individual relationships between temporal DE genes from cluster 1 (IFN-signaling) and the top 5 significantly enriched biological processes. Enriched pathway hubs (colored by biological process) were adjusted in size to reflect the number of genes associated with

each respective pathway, whereas for individual genes, their relationship is colored by the respective biological process (**a**). Bar graph illustrating the mean normalized expression levels over time for *IFNL1*, *IFNL2*, *IFNL3*, and *IFNB1*, at the respective temperatures for uninfected (Mock) hAEC cultures (top two panels) and for SARS-CoV (middle two panels) and SARS-CoV-2 (bottom two panels) infected hAEC cultures. Bars were adjusted in color to illustrate the respective SD among donors (**b**). The data underlying this figure are found in S1 Data. DE, differentially expressed; hAEC, human airway epithelial cell; IFN, interferon; SARS-CoV, Severe Acute Respiratory Syndrome Coronavirus; SARS-CoV-2, Severe Acute Respiratory Syndrome Coronavirus 2; SD, standard deviation.

viral replication and virus–host dynamics, particularly innate immune responses, during SARS-CoV and SARS-CoV-2 infection in human airway epithelial cells. Using an authentic in vitro model for the human respiratory epithelium, we demonstrate that SARS-CoV-2, in contrast to SARS-CoV, replicated up to 10-fold more efficiently at temperatures encountered in the upper respiratory tract. Concordantly, significantly increased amounts of Nucleocapsid-antigen positive cells were detected in these conditions. In addition, and despite intrinsic donor-to-donor variations, SARS-CoV-2 and SARS-CoV were both highly sensitive to pre-treatment with exogenous type I and type III IFNs. Importantly, a time-resolved transcriptome analysis showed temperature- and virus-dependent induction of the IFN-mediated antiviral and pro-inflammatory responses upon SARS-CoV and SARS-CoV-2 infections. Especially for SARS-CoV-2, the delayed triggering of innate immune responses coincided with the increased replication efficiencies at temperatures encountered in the upper respiratory tract.

One of the most profound phenotypical characteristics of fulminant SARS-CoV-2 is the early replication in the upper respiratory tract of infected individuals, which might facilitate the high transmissibility of SARS-CoV-2 [14,24,25,32]. In contrast, SARS-CoV was shown to primarily replicate in the lower respiratory tract and efficient transmissibility occurred at later stages of the clinical course [10,12,13]. Additionally, SARS-CoV was shown to poorly induce interferon and pro-inflammatory responses in infected cells [33,34]. The data presented here are in line with these features of SARS-CoV and SARS-CoV-2 infections and contribute to the understanding of the disparate human-to-human transmission dynamics for both zoonotic coronaviruses. They provide a framework to address the parameters of the molecular basis of exacerbations induced by SARS-CoV-2 infection in predisposed individuals.

Given that the receptor-binding motifs interacting with the human receptor ACE2 of the Spike proteins of both SARS-CoV and SARS-CoV-2 are highly conserved and that both SARS-CoV and SARS-CoV-2 displayed a similar cell tropism, the 380 amino acid differences, distributed across the entire genome and distinguishing SARS-CoV-2 from SARS-CoV, might account for specific interplays with host factors and differential replication efficiencies [2,6–9]. Another factor that may influence the temperature-dependent replication phenotype is the different form of the *ORF8* gene in SARS-CoV Frankfurt-1. The 29-nucleotide deleterious truncation in the *ORF8*, which is associated with a loss of fitness, was acquired during the initial human-to-human transmissions and was maintained in the SARS-CoV lineage that is at the origin of the international spread of SARS-CoV [35]. Therefore, besides comparing the replication of different SARS-CoV *ORF8* variants at 33˚C and 37˚C, it would be equally compelling to assess the phenotypic influence of similar truncations in the *ORF8* gene of SARS-CoV-2, especially since several SARS-CoV-2 isolates bearing a 382-nucleotide deletion truncating the *ORF8* gene have been detected [36]. Such SARS-CoV-2 *ORF8* variants can be readily engineered using the reverse genetic systems that were recently established for SARS-CoV-2 [32,37,38].

In this study, we report different temperature-dependent viral replication efficiencies for SARS-CoV and SARS-CoV-2, inversely associated with the amplitude of the innate immune response, albeit with a more pronounced phenotype for SARS-CoV-2. Our data are based on

the analysis of differentiated primary airway epithelial cell cultures derived from multiple donors, which were infected with SARS-CoV and SARS-CoV-2 for up to 96 hours. The substantial induction of interferon and pro-inflammatory responses in the airway epithelium following SARS-CoV-2 infection at 37˚C is consistent with other reports across multiple model systems, including undifferentiated primary cell-based systems analyzed between 24 and 48 hpi [39], stem cell-derived alveolospheres at 48 hpi [40], as well as with recent single-cell sequencing experiments performed on patient-derived samples [41]. An additional study concerned with transcriptomic analysis of patient-derived nasopharyngeal swabs found signs of a strong antiviral response and up-regulated chemokines, which was dependent on the viral load [42]. These findings were corroborated with primary airway cultures; however, compared to our own analysis, this interferon response was delayed. This difference might be explained by a divergent experimental setup. In contrast to other studies, we directly compared viral replication and host responses at 33˚C and 37˚C, and report delayed innate immune and pro-inflammatory pathways activation at 33˚C. Furthermore, a recently performed genome-wide CRISPR screen showed that SARS-CoV-2 replication requires partially different host factors when incubated at either 33˚C or 37˚C [43,44].

Foxman and colleagues elegantly described, in an analogous model to the hAEC cultures and by using common cold viruses, that the PRR-mediated IFN response is influenced by temperature [23]. This may also apply in the context of SARS-CoV-2 infections; however, due to the multifaceted intricate nature of virus–host interactions, it is likely that the efficient replication of SARS-CoV-2 and the concurrent expression of a plethora of known coronavirus-encoded factors that antagonize host antiviral response also play a crucial role herein [45–50]. Nonetheless, we demonstrate that SARS-CoV-2 and SARS-CoV are highly sensitive to both type I and III IFN-driven responses. These data are supported by several studies investigating the outcome of IFN pretreatment in cultured cell lines [39,51,52], as well as the well-documented dominant antiviral role of type III IFN during virus infection in the respiratory epithelium [31,53,54]. IFN lambda therefore represents an attractive candidate for the development of intervention strategies against SARS-CoV-2 respiratory infections.

The detailed replication dynamics of both SARS-CoV and SARS-CoV-2 in hAEC cultures, as well as the time-resolved host responses to infection reported here provide crucial insight into the profound impact of ambient temperatures on pivotal virus–host interactions in the airway epithelium [55]. These data will likely be extended by additional mechanistic and functional in vivo studies delineating the efficacy of antiviral host responses triggered by SARS-CoV and SARS-CoV-2 infections, as well as deciphering the influence of virus-encoded antagonists and physical parameters. This knowledge should be exploited broadly to support clinical applications to combat SARS-CoV-2 infections.

## Methods

### Cells and human airway epithelial cell (hAEC) cultures

Vero E6 cells (kindly provided by Doreen Muth, Marcel Müller, and Christian Drosten, Charité, Berlin, Germany) were propagated in Dulbecco's Modified Eagle Medium-GlutaMAX supplemented with 1 mM sodium pyruvate, 10% (v/v) heat-inactivated fetal bovine serum (FBS), 100 µg/ml streptomycin, 100 IU/ml penicillin, 1% (w/v) nonessential amino acids, and 15 mM HEPES (Gibco, Gaithersburg, Maryland, United States of America). Cells were maintained at 37˚C in a humidified incubator with 5% $CO_2$.

Primary human tracheobronchial epithelial cells were isolated from patients (>18 years old) undergoing bronchoscopy or pulmonary resection at the Cantonal Hospital in St. Gallen, Switzerland, or Inselspital in Bern, Switzerland, in accordance with ethical approval (EKSG

11/044, EKSG 11/103, KEK-BE 302/2015, and KEK-BE 1571/2019). Isolation and culturing of primary material was performed as previously described [56]. Briefly, cryopreserved cells were thawed and expanded for 1 week in BEGM medium. After initial expansion phase, cells were transferred into in collagen type IV-coated porous inserts (6.5 mm radius insert, Costar, Corning, New York, USA) in 24-well plates. Cells were expanded for another 2 to 3 days in BEGM in a liquid–liquid state. Once the cells reached confluency, the basolateral medium was exchanged for air-liquid interface (ALI) medium, and the apical medium was removed to allow for the establishment of the ALI. Basolateral ALI medium was exchanged 3 times per week, and apical side was washed with Hanks balanced salt solution (HBSS, Gibco) once a week, until the development of a fully differentiated epithelium (3 to 4 weeks), which was monitored by optical microscopy. Several modifications to the original protocol were used. The concentrations of hydrocortisone for both BEGM and ALI were increased to 0.48 μg/ml, and BEGM was further supplemented with the inhibitors 1 μmol/L A83-01 (Tocris, USA), 3 μmol/L isoproterenol (Abcam, Cambridge, United Kingdom), and 5 μmol/L Y27832 (Tocris, USA) [57]. Basolateral ALI medium was exchanged 3 times per week, and apical side was washed with HBSS (Gibco) once a week. hAEC cultures were maintained at 37˚C in a humidified incubator with 5% $CO_2$.

## Viruses

SARS-CoV strain Frankfurt-1 (GenBank FJ429166) [35,58] and SARS-CoV-2 (SARS-CoV-2/München-1.1/2020/929) [37] were kindly provided by Daniela Niemeyer, Marcel Müller, and Christian Drosten, and propagated and titrated on Vero E6 cells.

## Infection of hAEC cultures

Well-differentiated hAEC cultures were infected with 30,000 plaque-forming units (PFU) of either SARS-CoV or SARS-CoV-2. Viruses were diluted in HBSS (Gibco), inoculated on the apical side, and incubated for 1 hour at either 33˚C or 37˚C. Afterwards, virus inoculum was removed, and the apical surface washed 3 times with HBSS, whereby the third wash was collected as the 1 hpi time point. The cells were incubated at the indicated temperatures in a humidified incubator with 5% $CO_2$. Released virus progeny were monitored every 24 hours by incubating 100 μl of HBSS on the apical surface 10 minutes prior to the time point. The apical washes were collected, diluted 1:1 with virus transport medium (VTM), and stored at −80˚C for later analysis. Basolateral medium was collected at each time point and stored at −80˚C for later analysis. Fresh ALI medium was then added to the basolateral compartment. To analyze virus replication following IFN exposure, hAEC cultures were pretreated with recombinant universal type I IFN (100 or 10 IU/ml; Sigma Aldrich, Buchs, St. Gallen, Switzerland) or recombinant IFN-λ3 (100 or 10 ng/ml [59]) for 18 hours from the basolateral side, prior to infection and incubated at either 33˚C or 37˚C. As controls, untreated hAEC cultures were used. Shortly before infection with SARS-CoV and SARS-CoV-2, the basolateral medium containing type I or type III IFN was removed and replaced with medium without exogenous IFN.

## Immunofluorescence analysis of infected hAECs

Well-differentiated hAEC cultures were fixed with 4% (v/v) neutral buffered formalin and processed as previously described [56]. Cells were permeabilized in PBS supplemented with 50 mM $NH_4Cl$, 0.1% (w/v) Saponin, and 2% (w/v) bovine serum albumin (CB). To detect SARS-CoV and SARS-CoV-2, hAEC cultures were immunostained with a rabbit polyclonal antibody against SARS-CoV Nucleocapsid protein (Rockland, 200-401-A50), which also cross-react with SARS-CoV-2. Cell distribution of ACE2 were detected with a rabbit polyclonal antibody

against ACE2 (ab15348, Abcam). Alexa Fluor 488-labeled donkey anti-rabbit IgG (H + L) (Jackson Immunoresearch, Cambridgeshire, UK) was used as secondary antibody. Alexa Fluor 647-labeled rabbit anti-β-tubulin (9F3, Cell Signaling Technology, Danvers, Massachusetts, USA) and Alexa Fluor 594-labeled mouse anti-ZO1 (1A12, Thermo Fisher Scientific, Darmstadt, Germany) were used to visualize cilia and tight junctions, respectively. Antibodies were diluted in CB. All samples were counterstained using 4′,6-diamidino-2-phenylindole (DAPI, Thermo Fisher Scientific) to visualize the nuclei. Samples were imaged on a DeltaVision Elite High-Resolution imaging system (GE Healthcare Life Sciences, Chicago, Illinois, USA) equipped with 60x oil immersion objective (1.4 NA), by acquiring 200 to 300 nm z-stacks over the entire thickness of the sample. Images were deconvolved using the integrated softWoRx software. For the quantification of infected cells, images were alternatively acquired using an EVOS FL Auto 2 Imaging System equipped with a 20x air objective. All images were processed using FIJI software packages [60]. Brightness and contrast were adjusted identically to their corresponding controls. Figures were assembled using the FigureJ plugin [61]. Quantification of infected cells from 4 donors was performed by morphological segmentation of individual cells using the ZO-1 staining and the MorphoLibJ plugin in FIJI [62]. Each region of interest was used to measure the mean intensity in the channel corresponding to the nucleocapsid staining. Cells with mean intensities > mean + 3 standard deviations compared to the distribution of mock-infected cells were considered positive. On average, over $10^4$ cells were analyzed per donor and per condition.

## Titration of apical and basolateral compartments

Viruses released into the apical or basolateral compartments were titrated by plaque assay on Vero E6 cells. Briefly, 1.7 * $10^5$ cells/ml were seeded in 24-well plates 1 day prior to the titration and inoculated with 10-fold serial dilutions of virus solutions. Inoculums were removed 1.5 hpi and replaced with overlay medium consisting of DMEM supplemented with 1.2% Avicel (RC-581, FMC biopolymer), 5% heat-inactivated FBS, 50 μg/ml streptomycin, and 50 IU/ml penicillin. Cells were incubated at 37°C 5% $CO_2$ for 48 hours, fixed with 4% (v/v) neutral buffered formalin, and stained with crystal violet.

## Bulk RNA Barcoding and sequencing (BRB-seq) and data analysis

Total cellular RNA from mock and virus-infected hAEC cultures was extracted using the NucleoMag RNA kit (Macherey-Nagel, Oensingen, Switzerland) according to the manufacturer's guidelines on a Kingfisher Flex Purification system (Thermofisher). Total RNA concentration was quantified with the QuantiFluor RNA System (Promega, Madison, WI, USA) according to the manufacturer's guidelines on a Cytation 5 multimode reader (Biotek, Sursee, Switzerland). A total of 100 ng of total cellular RNA was used for the generation of BRB-seq libraries, and subsequent sequencing on an Illumina HiSeq 4000 platform was performed as described previously to a depth of approximately 12 million raw reads per sample [30]. The sequencing reads were demultiplexed using the BRB-seqTools suite and were aligned against a concatenation of the human genome (hg38), the SARS coronavirus Frankfurt-1 (AY291315) genome, and the SARS-CoV-2/Wuhan-Hu1/2020 (NC_045512) genome using the STAR aligner and HTSeq for producing the count matrices [30,63,64]. Following the alignment, the raw count matrices were randomly downsampled across all samples to compute the sequence saturation using the average number of reads per sample and median number of detected genes (>1 read) across samples. All downstream analyses were performed using R (version 3.6.1). ComBat-seq was used with default settings to adjust for batch effects in the raw data and generate an adjusted count matrix used for downstream analyses [65]. Library

normalization and expression differences between uninfected and virus-infected samples were quantified using the DESeq2 package (version 1.28) with a fold change (FC) cut-off of $\geq 1.5$ and a false discovery rate (FDR) of $\leq 0.1$ [66]. Due to the multifactor design of these experiments, DE analysis was performed using several approaches: (1) Samples were subset by temperature prior to DE analysis (e.g., subset of samples for all time points at 33°C), and infected samples were compared to uninfected samples using the design ~ Batch + Condition; (2) Samples were subset by temperature and time prior to DE analysis (e.g., subset of samples at 33°C and 24 hpi), and infected samples were compared to uninfected samples using the design ~ Batch + Condition; (3) For the temporal analysis, all samples were kept together, and the identification of significant DE genes over time was performed using the likelihood ratio test (LRT) with both the complete design ~ Condition + TH + Condition:TH (TH = conjugation of the Temperature and Time variables) and the reduced design ~ Condition + TH. Hierarchical gene clustering was subsequently performed on a variance-stabilizing transformation (VST) processed count matrix of identified DE genes using the degPatterns function from the DEGreports package [67]. Venn diagrams of overlapping DE genes were generated using the VennDiagram package [68]. Pathway enrichment analysis was performed using the clusterProfiler and ReactomePA packages in R [69,70]. Significantly enriched pathways with a gene count >1 and *p*-value of ≤0.05 were visualized using the enrichplot package. Further data analysis and visualization was performed using a variety of additional packages in R, including ComplexHeatmap and ggplot2 [71].

## Bioinformatic analysis of ACE2 and TMPRSS2 expression

For the analysis of *ACE2*, *TMPRSS2*, *IFNAR1*, *IFNAR2*, *IFNRL1*, and *IL10RB* mRNA expression, we reanalyzed previously obtained single-cell raw sequencing data from uninfected hAEC cultures [31]. The resulting unique molecule identifier (UMI) count matrix for each sample was preprocessed and filtered individually, and then, samples were merged in Seurat (v3.1) [72]. Data scaling, normalization, and regression of unwanted sources of variation (number of UMIs, percentage of mitochondrial reads, cell cycle phase) were performed using the integrated SCtransform option in Seurat, followed by dimensional reduction using UMAP (Uniform Manifold Approximation and Projection) embedding. For cell type annotation, the resulting integrated dataset was used for unsupervised graph-based clustering to annotate the different cell types using both cluster-specific marker genes and well-known canonical marker genes to match the identified clusters with specific cell types found in the respiratory epithelium, as described previously [31].

## Statistical testing

Distribution testing was performed using the Shapiro–Wilk normality test (>0.05), followed by computing the *P* value of the mean log10 PFU/ml at each time point or treatment between SARS-CoV and SARS-CoV-2 using a two-sided paired sample *t* test. Analyses were performed using R (version 3.6.1) or SciPy using Python (Version 3.7).

## Supporting information

**S1 Fig. Basolateral release of SARS-CoV and SARS-CoV-2 in infected hAEC cultures.** Well-differentiated hAEC cultures were infected with SARS-CoV and SARS-CoV-2 using 30,000 PFU or remain uninfected (mock) and were incubated at 37°C (**a**) or 33°C (**b**). Inoculated virus was removed at 1 hpi, and the apical side was washed. Cultures were further incubated at the indicated temperature. At the indicated time post infection, virus release in the basolateral compartment was assessed by plaque titration (**a, b**). Data represent the mean 2

two replicates. The data underlying this figure are found in S1 Data. hAEC, human airway epithelial cell; hpi, hours post infection; PFU, plaque-forming unit; SARS-CoV, Severe Acute Respiratory Syndrome Coronavirus; SARS-CoV-2, Severe Acute Respiratory Syndrome Coronavirus 2.
(TIF)

**S2 Fig. ACE2 is expressed on both ciliated and nonciliated cells.** Immunofluorescence analysis of ACE2 receptor distribution in unexposed well-differentiated hAEC cultures (**a**). Unexposed well-differentiated hAEC cultures were fixed and processed for microscopy analysis using antibodies against ACEC2 (green), β-tubulin (cilia, red), ZO-1 (tight junctions, white), and DAPI (blue). Representative z-projections of 3 donors are shown. Scale bar, 20 μm. Unexposed well-differentiated hAEC cultures from different human donors were used to perform scRNA-seq analysis. The UMPA plots shows the dimensional reduction of the 8,128 cells belonging to either basal, goblet, secretory, preciliated, and ciliated cell population (**b**), as well as the relative expression level and distribution of ACE2 (**c**) and TMPRSS2 (**d**), respectively. In addition, the respective relative expression level and distribution of both the IFN-alpha/beta receptor alpha and beta chain, IFNAR1 (**e**), IFNAR2 (**f**), and the type III IFN receptor complex IFNLR1 (**g**), and IL-10RB (**h**). ACE2, angiotensin-converting enzyme 2; hAEC, human airway epithelial cell; IFN, interferon; scRNA-seq, single-cell RNA-sequencing; UMAP, Uniform Manifold Approximation and Projection.
(TIF)

**S3 Fig. Sequence saturation plots of BRB-seq libraries.** Sequence saturation graphs from 2 individual BRB-seq libraries, encompassing 72 (**a**) or 96 (**b**) samples from 3 or 4 independent biological donors, respectively, illustrating the average number of reads (x-axis) and median number of detected genes per library (y-axis; >1 read) across samples (left) and total sequencing depth (right; M = Million). BRB-seq, Bulk RNA Barcoding and sequencing.
(TIF)

**S4 Fig. Percentage viral reads in untreated and virus-infected hAEC cultures.** Boxplots illustrating the distribution of the fraction of SARS-CoV (**a**) and SARS-CoV-2 (**b**) specific viral reads among the total fraction of sequence reads at the different time points and temperature conditions in the respective individual untreated (Mock), SARS-CoV, and SARS-CoV-2 samples (colored by donor). hAEC, human airway epithelial cell; SARS-CoV, Severe Acute Respiratory Syndrome Coronavirus; SARS-CoV-2, Severe Acute Respiratory Syndrome Coronavirus 2.
(TIF)

**S5 Fig. Overlap of differentially expressed genes in SARS-CoV and SARS-CoV versus SARS-CoV-2 infected hAEC cultures.** Venn diagrams showing the overlap of DEG in SARS-CoV or SARS-CoV-2 infected hAEC cultures among the different time point and temperature conditions (**a-d**), and SARS-CoV-2 contrasted to SARS-CoV at the different time point and temperature conditions (**e, f**). DEG, differentially expressed genes; hAEC, human airway epithelial cell; SARS-CoV, Severe Acute Respiratory Syndrome Coronavirus; SARS-CoV-2, Severe Acute Respiratory Syndrome Coronavirus 2.
(TIF)

**S6 Fig. Heatmap of mean expression levels of temporal differentially expressed genes.** A heatmap illustrating the hierarchical clustering of the expression levels of 401 unique DE genes identified in the pairwise comparison of SARS-CoV and SARS-CoV-2 infected hAEC cultures compared to uninfected hAEC cultures (Mock) at different time points and incubation

temperatures. Expression levels for individual temporal DE genes are shown in rows as the $\log_2$ mean normalized counts for 7 human donors stratified by condition, temperature, and hours post infection (columns; representative colors shown in legends). The 29 temporal DE genes unique for cluster 1 are shown on the right (y-axis). DE, differentially expressed; hAEC, human airway epithelial cell; SARS-CoV, Severe Acute Respiratory Syndrome Coronavirus; SARS-CoV-2, Severe Acute Respiratory Syndrome Coronavirus 2.
(TIF)

**S7 Fig. Chemokine and cytokine expression in untreated and virus-infected hAEC cultures.** Bar graph illustrating the mean normalized expression levels over time for the cytokines IL11, IL18, IL1b, and TNF (**a**), or the chemokines CCL2, CCL5, CXCL10, and CXCL11 (**b**), at the respective temperatures for Mock (uninfected) hAEC cultures (top two panels) and for SARS-CoV (middle two panels) and SARS-CoV-2 (bottom two panels) infected hAEC cultures. Bars were adjusted in color to illustrate the respective SD among donors. hAEC, human airway epithelial cell; SARS-CoV, Severe Acute Respiratory Syndrome Coronavirus; SARS-CoV-2, Severe Acute Respiratory Syndrome Coronavirus 2; SD, standard deviation.
(TIF)

**S1 Table. Differentially expressed genes in SARS-CoV and SARS-CoV-2 infected hAEC cultures relative to uninfected hAEC cultures at 33˚C or 37˚C.**
(XLSX)

**S2 Table. Overlap among differentially expressed genes identified in SARS-CoV and SARS-CoV-2 -infected hAEC cultures at either 33˚C or 37˚C.**
(XLSX)

**S3 Table. Differentially expressed genes in SARS-CoV and SARS-CoV-2 infected hAEC cultures relative to their corresponding uninfected samples at each temperature and time point.**
(XLSX)

**S4 Table. Differentially expressed genes in SARS-CoV-2 infected hAEC cultures relative to SARS-CoV infected hAEC cultures at each temperature and time point.**
(XLSX)

**S5 Table. Differentially expressed genes associated with hierarchical gene clusters 1, 2, and 3.**
(XLSX)

**S1 Data. Raw data underlying Figs 1A, 1B, 1D, 2A, 2B, 4B, and S1A and S1B Fig.**
(XLSX)

## Acknowledgments

We gratefully thank the École Polytechnique Fédérale de Lausanne (EPFL) and the University of Bern for providing special authorization to conduct our research during the SARS-CoV-2 outbreak. We are grateful to Sabina Berezowska and Irene Ramos-Centeno (Institute of Pathology, University of Bern) for providing the tissues via the Tissue Bank Bern.

## Author Contributions

**Conceptualization:** Ronald Dijkman.

**Data curation:** Philip V'kovski, Mitra Gultom, Jenna N. Kelly, Silvio Steiner, Ronald Dijkman.

**Formal analysis:** Philip V'kovski, Mitra Gultom, Jenna N. Kelly, Silvio Steiner, Ronald Dijkman.

**Funding acquisition:** Volker Thiel, Ronald Dijkman.

**Investigation:** Philip V'kovski, Mitra Gultom, Jenna N. Kelly, Silvio Steiner, Julie Russeil, Bastien Mangeat, Elisa Cora, Joern Pezoldt, Melle Holwerda, Annika Kratzel, Laura Laloli, Manon Wider, Jasmine Portmann, Thao Tran, Nadine Ebert, Hanspeter Stalder, Ronald Dijkman.

**Methodology:** Philip V'kovski, Ronald Dijkman.

**Project administration:** Ronald Dijkman.

**Resources:** Rune Hartmann.

**Supervision:** Vincent Gardeux, Daniel Alpern, Bart Deplancke, Volker Thiel, Ronald Dijkman.

**Validation:** Philip V'kovski, Mitra Gultom, Jenna N. Kelly, Silvio Steiner, Ronald Dijkman.

**Visualization:** Philip V'kovski, Mitra Gultom, Jenna N. Kelly, Silvio Steiner, Ronald Dijkman.

**Writing – original draft:** Philip V'kovski, Ronald Dijkman.

**Writing – review & editing:** Philip V'kovski, Mitra Gultom, Jenna N. Kelly, Silvio Steiner, Volker Thiel, Ronald Dijkman.

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
