## [Editor Report · Decision Letter 0]

22 Jan 2021

Dear Ronald, 

Thank you for submitting your manuscript entitled "Disparate temperature-dependent virus – host dynamics for SARS-CoV-2 and SARS-CoV in the human respiratory epithelium" for consideration as a Research Article by PLOS Biology.

Your manuscript has now been evaluated by the PLOS Biology editorial staff, as well as by an academic editor with relevant expertise, and I am writing to let you know that we would like to move forward with your manuscript and that we want to make a quick decision, trying to avoid going to an arbitrating reviewer but we may need to. However, before we need you to complete your submission by providing the metadata that is required for full assessment. To this end, please login to Editorial Manager where you will find the paper in the 'Submissions Needing Revisions' folder on your homepage. Please click 'Revise Submission' from the Action Links and complete all additional questions in the submission questionnaire. 

Please re-submit your manuscript within two working days, i.e. by Jan 24 2021 11:59PM.

Kind regards,

Paula

---

Associate Editor

PLOS Biology

---

## [Editor Report · Decision Letter 1]

11 Feb 2021

Dear Dr Dijkman,

Thank you very much for submitting your manuscript "Disparate temperature-dependent virus – host dynamics for SARS-CoV-2 and SARS-CoV in the human respiratory epithelium" for consideration as a Research Article by PLOS Biology. Your manuscript was evaluated by the PLOS Biology editors as well as by an Academic Editor with relevant expertise and we all appreciated the attention to an important topic. 

We will probably accept this manuscript for publication, provided you satisfactorily address the following data and other policy-related requests.

DATA POLICY:

Regardless of the method selected, please ensure that you provide the individual numerical values that underlie the summary data displayed in the following figure panels as they are essential for readers to assess your analysis and to reproduce it: Figure 1A, 1B, 1D, 2A, 2B, 4B, Supplementary Figure 1A, and 1B.

**Please also ensure that figure legends in your manuscript include information on where the underlying data can be found**, and ensure your supplemental data file/s has a legend.

We expect to receive your revised manuscript within two weeks.

*Published Peer Review History*

*Early Version*

Sincerely,

Paula

---

Associate Editor,

pjaureguionieva@plos.org,

PLOS Biology

---

## [Editor Report · Decision Letter 2]

25 Feb 2021

Dear Dr. Dijkman,

On behalf of my colleagues and the Academic Editor, Ken Cadwell, I am pleased to say that we can in principle offer to publish your Research Article "Disparate temperature-dependent virus – host dynamics for SARS-CoV-2 and SARS-CoV in the human respiratory epithelium" in PLOS Biology, provided you address any remaining formatting and reporting issues, if any. These will be detailed in an email that will follow this letter and that you will usually receive within 2-3 business days, during which time no action is required from you. Please note that we will not be able to formally accept your manuscript and schedule it for publication until you have made the required changes.

PRESS

Thank you again for supporting Open Access publishing. We look forward to publishing your paper in PLOS Biology. 

Sincerely, 

Paula

---

Paula Jauregui, PhD 

Senior Editor 

PLOS Biology